# Psychometric properties of the Arabic version of the Existence Scale

**Othman A. Alfuqaha**[1]*, **Mohammed M. Al-Hammouri**[2], **Jehad A. Rababah**[2], **Bayan A. Alfoqha**[3], **Ola N. Alfuqaha**[4], **Moh'd Fayeq F. Haha**[5], **Suzan S. Musa**[5], **Aseel A. Matter**[6]

1 Department of Nursing, Jordan University Hospital, The University of Jordan, Amman, Jordan, 2 School of Nursing, Jordan University of Science & Technology, Irbid, Jordan, 3 Ministry of Education, Amman, Jordan, 4 School of Nursing, Hashemite University, Zarqa, Jordan, 5 School of Medicine, Jordan University Hospital, The University of Jordan, Amman, Jordan, 6 School of international studies, Ewha Womans University, Seoul, Republic of South Korea

* Othman_alfoqaha@yahoo.com

**Data Availability Statement:** The data underlying the results presented in the study are available on the following link: https://figshare.com/articles/dataset/Psychometric_properties_sav/19219923.

## Abstract

The Existence Scale (ES) is a theory-based measure assessing personal fulfillment and finding meaning in life. This study aims to translate the ES into Arabic language and test its psychometric properties in Jordan populations. A methodological design was performed on a convenience sample of 551 participants by three samples of nurses, schoolteachers, and undergraduate students. Data collection was carried out between February and May 2019. Translation and back translation, face validity (Important Score>1.5), content validity ratio (CVR>0.62) and index (CVI/Ave>0.80), construct, convergent, and discriminant validity were obtained. Furthermore, Cronbach's alpha, composite reliability, and average variance extracted were investigated in this study. The results showed that five items were deleted based on content validity ratio and four items were deleted based on their low factor loading. The exploratory factor analysis showed four subscales for the translated ES (37 items), explaining 61.57% of the variance collectively. The confirmatory factor analysis supported the four subscales with acceptable goodness of fit indices. The result of the total Cronbach's alpha for the ES was 0.93, and for subscales it ranged from 0.88 to 0.93. Composite reliability and average variance extracted results for the translated ES were supportive of the reliability. These results confirm that the translated Arabic version of the ES (37 items) in Jordan populations is acceptable regarding validity and reliability.

## Introduction

Existential approach currently draws the attention of many scientists in psychology, nursing, and related fields. The existential theory stresses the free and responsible dimension in life under which human beings have the capability of making choices and decisions in life based on their internal value system that guides the decision making to find a meaningful existence [1]. Search for meaning is an important goal in life, as a result, the absence of meaning in life, which is called "Existential Vacuum" causes many psychological problems such as burnout [2, 3], depression [4], and substance abuse [5].

**Funding:** The author(s) received no specific funding for this work.

**Competing interests:** The authors have declared that no competing interests exist.

Scales to measure fullness of existence, purpose, and meaning in life have been developed such as The Multidimensional Existential Meaning Scale (MEMS) [6], Meaning in Life Questionnaire (MLQ) [7], and Purpose in Life Scale (PLS) [8]. Among them, the Existence Scale (ES) has been used far more than any other global measure in fullness of existence [9, 10]. ES has been recognized as an important self-rating instrument to identify the principal dimension in human personality, measure psychological problems, and assess personal fulfillment and find meaning in life [1]. For these benefits, clarifying the existence among individuals may need to be jointly prioritized in the treatment of existential vacuum and other psychological issues. In this regard, the ES is a reliable and valid measure in various populations and languages [2, 10, 11].

Langle et al. [1] who developed the ES, consisted of 46 items divided into four subscales namely self-distance, self-transcendence, freedom, and responsibility. In addition to its use in assessing personal fulfillment and finding meaning in life, the ES has been used to evaluate psychotherapeutic interventions' efficiency [12]. For example, in Russia (2020), they found positive effects of psychotherapeutic experience on all aspects of personal fulfillment: self-distance, self-transcendence, freedom, and responsibility. Besides, healthy narcissistic self-regulation was associated with personal fulfillment, and both were associated with the stability and integrity of the self-system in appreciating one's personality and finding a meaningful life [13].

ES has been argued to be an essential aspect of any professional activity to energetically promote aspects of the personality's functioning [14]. In this regard, nursing, teaching, and student professions are considered stressful professions that lead to much psychological distress such as burnout, sense of emptiness, low self-evaluation, and negative affect of meaning in life. This makes them at high risk for existential vacuum [14–16].

In a number of studies, performed in different countries such as Russia, Hungary, and Iran, ES has been found acceptable and valid measure of existential meaning in life [2, 10, 17]. In contrast, a study performed in four different samples in Netherlands has failed to confirm the validity and reliability of the ES [11]. Thus, further studies are needed to measure the validity and reliability of ES in different samples and various populations. Of note, the meaning in life among social professions (i.e., nursing) that have extensive contact with people not only affects the professionals themselves but is also affected the quality-of-life people [2, 3].

Despite the evidence in the literature on the importance and application of the ES in research and its association with various health, business, and personal variables in various populations, the scale was not used in the Arabic context. Moreover, one of the reasons for overlooking such an important scale could be the lack of valid and reliable scales to assess meaning and purpose in life in Jordan populations. Our argument is supported by inability to locate a single article about translation and validation of the ES in Jordan populations.

In conclusion, there is a need to further investigate and psychometric properties of the ES and provide translated version to be used in Jordan populations. Thus, the current study's goal was to translate the ES into Arabic language and test its psychometric properties in the recruited areas in the Jordan population. The recruited areas (nursing, teaching, and students) were selected based on the evidence that experience extensive contact with people and high level of stress [11]. In addition, the recruited samples are expected to improve the generalizability of the findings.

## Materials and methods

### Study design

A methodological design was used to test psychometric properties of the ES in Jordan populations. Data collection was carried out between February and May 2019. The protocol for the translation and psychometric properties testing are discussed below.

**Description of the original ES.** The official permission was sought from the developer (Alfried Längle) to translate, modify, and omit the ES as needed. However, the original ES by Langle and his colleagues (2003) consists of 46 items divided into four subscales. The first subscale is self-distance (SDi), consists of 8 questions assess an event experience. The second subscale is self-transcendence (STr) subscale consists of 14 questions assess the inner spiritual essence of life, awareness, and the true meaning of life. The third subscale is freedom (Fr) subscale, consists of 11 questions assess the ability to choose their attitude. The fourth subscale is responsibility (Re) subscale consisting of 13 questions assess the responsibility of decisions and tasks. All items in the ES were rated on a 5-point Likert-type scales as follows: 1 'Absolutely', 2 'Almost', 3 'Not Really', 4 'No, hardly', 5 'Not at all'. Items number (9, 13, 17, 21, and 26) were rated on a reversed order. The score of each item was counted in every construct of the ES. Higher scores indicate existential fulfillment [1].

## Translation phase

The translation process of the original ES was examined by 4 bilingual independent experts (PhD holder specialized in English-Arabic translation, nursing, and psychology) throughout two steps: translation and back translation taking into consideration the original term, and translating it in the relevant term, simple and clear, and cultural equivalence in fewer words [18]. Two bilingual independent experts (English-Arabic language) did the forward translations into Arabic language then they agreed on the combined single version. After that, another two independent experts (PhD holder specialized in nursing and psychology), did the reverse translations into English language.

## Validity

To determine the validity, face, content, construct, convergent, and discriminant validity were used and illustrated as follow:

**Face validity.** To evaluate the quantitative face validity, a pilot study with 10 participants from nurses was done, we asked them to provide their suggestions regarding context, linguistic, ambiguity, and simplicity of the translated version of the ES items. Therefore, importance scores equal or larger than 1.5 were considered for analysis in this study [19].

**Content validity.** Content validity was performed with the participation of 10 arbitrators as follows: 3 PhD holder specialized in nursing, 3 PhD holder in psychology, 2 PhD holder in psychiatry, and 2 PhD holder in public health. They were requested to provide their suggestions regarding context, linguistic, and suitability for the local society and scored based on a 3-point Likert scale for content validity ratio (CVR), content validity index (CVI), and Kappa value.

Regarding CVR, the acceptance rate for 10 arbitrators was at least 0.62 based on Lawshe's CVR Table for all items [20]. CVI was assessed through the average for all items from the arbitrators on the number of items, if the item was scoring less than 0.80 should be omitted from the translated ES [21]. Kappa value above 0.70 for each item in the translated ES was considered superior [21].

**Construct validity.** To determine the construct validity, exploratory factor analysis (EFA) was conducted on the translated ES. The following criteria were used to evaluate the results of the EFA: 1- principal component analysis with varimax rotation set at 0.30 for all items [22, 23], 2- Eigenvalues greater than 1.5 are represented factor [24], 3- sampling adequacy by using Kaiser-Meyer-Olken (KMO) test greater than 0.70 is considered adequate [25], and 4- Bartlett's test of sphericity $p < 0.05$ is deemed significant [26].

However, the three samples consisted of three professions (nursing, teaching, and students) that require social occupations, extensive contact with people, and more strenuous job than others. Besides, selection of diverse population will improve the generalizability of our findings. We selected the three samples population based on previous studies [11, 17] to perform EFA and confirmatory factor analysis (CFA).

To allocate the sample size, a sample of 100–300 is recommended in every profession [27]. Trying to cover all sector in the selected profession, we divided the profession into two major sectors (i.e., public vs. private). We visited their respective target places and asked them to participate in this study. A purposeful sample of 300 participants was selected to represent every single profession (200 public vs 100 private). The returned valid questionnaires for analysis from nurses in public hospital were 112 participants with a response rate of 56%, while the returned valid questionnaires from nurse in private hospital were 65 participants with a response rate of 65%. The returned valid questionnaires from public school teachers were 106 participants with a response rate of 53%, while the returned valid questionnaires from private school teachers were 49 participants with a response rate of 49%. The returned valid questionnaires from public student university students were 152 participants with a response rate of 78%, while the returned valid questionnaires from private student university were 67 participants with a response rate of 67%. The completed questionnaire was entered and analyzed by the statistical package for the social science (SPSS) version 22.0 and analysis of moment structure (AMOS) version 26.0.

**Confirmatory factor analysis.** We considered the CFA depend on a maximum likelihood method in AMOS program to assess the factor structure of the translated ES scale and its subscales, the following criteria were applied as follows: 1- Factor loadings of 0.30 and greater would represent a fair contribution of latent variables [28], 2- Relative Chi-square ($\chi$2/df ratio) less than 3 is acceptable, 3- Root Mean Square of Error Approximation (RMSEA) value less than 0.05 indicate an excellent mode, 4- Comparative Fit Index (CFI) value greater than 0.95 will reveal the best model fit, 5- Goodness of Fit Index (GFI) value > 0.90 indicate an excellent model fit while the values between 0.85–0.90 are indicative of acceptable model, and 6- Increment Fit Index (IFI) and Tucker-Lewis Index (TLI) values > 0.90 are indicative of a good model fit [21, 29].

**Convergent and discriminant validity.** Convergent validity was evaluated by examining average variance extracted (AVE). The AVE had to be greater than 0.50 [30]. Discriminant validity was assessed by examining squared root of the AVE, maximum shared variance (MSV), and average shared squared variance (ASV). The square root of AVE should be greater than inter-factor correlations of ES (SDi, STr, Fr, and Re) [30]. Both MSV and ASV values had to be lower than AVE values [27].

## Reliability

To assess the reliability of the translated ES, the composite reliability (CR) and Cronbach's alpha were detected. If the value is greater than 0.70 in CR considered acceptable [31]. Also, Cronbach's alpha more than 0.80 indicates internal reliability [32].

## Ethical considerations

An envelope containing the consent form, demographic factors, and the translated ES was constructed. Then, the researchers in person began to distribute the envelop take into consideration their completely agreement to participate in this study. Ethical guidelines set by the institutional review board No. (10/2019/4140) at the selected hospital were followed during the collection of data. Moreover, written informed consent was obtained from each participant

in this study. Participants were assured of their right to refuse to participate in this study and confirmed that their responses will be confidential and anonymous.

## Results

### Translation phase

The translators met and agreed on the final translated of the ES. All translators met and discussed any inconsistencies until they reached 100% agreement on 46 items of ES.

### Face validity

With feedback from piloting participants, we made some amendments in terms of linguistics ambiguity, simplicity, and local suitability in the translated version of the ES. Accordingly, all items had achieved the score of equal or higher than 1.5; none of the items was removed from the translated ES in this stage.

### Content validity

For evaluation of the content validity, CVR was calculated, and the result showed that 5 items (4 items from self-transcendence and 1 item from freedom) were rated below 0.62 according to Lawshe's CVR Table, leaving the translated ES from 46 to 41 items. However, based on CVI, the result showed that the average score was equal to 0.91. The kappa value for each item was more than 0.70, hence, content validity of the translated ES was achieved on 41 items.

### Construct validity

**KMO Test.** The KMO test result was 0.91. This value is considered indicative of sampling adequacy and that factor analysis would yield reliable and distinct factors.

**Bartlett's test of sphericity.** The result shows typically significant and suitable for factor analysis of the 41 items of the translated ES among Jordanian participants in this study (Chi-square ($\chi$2): 20875.5; df: 820; P<0.001).

The frequencies and percentages were calculated for the demographic factors. The number of females in our sample was 338 (61.3%), 419 of the sample were $\leq$35 years (76%), and 219 were students (39.7%). Table 1. displays the demographic factors of the study sample.

EFA was conducted on 41 items of the translated ES among Jordanian samples of nurses, teachers, and undergraduate students. The results of the factor loading of the ES (41 items) are shown in Table 2.

**Table 1. Demographic characteristics of the study sample (n = 551).**

| Variable: | Nurses n (%) | Teachers n (%) | University Students n (%) |
|---|---|---|---|
| Gender | 177 | 155 | 219 |
| Male | 85 (48%) | 60 (38.7%) | 68 (31.1%) |
| Female | 92 (52%) | 95 (61.3%) | 151 (68.9%) |
| Age: | | | |
| $\leq$35 Years | 135 (76.3%) | 91 (58.7%) | 193 (88.1%) |
| >35 Years | 42 (23.7%) | 64 (41.3%) | 26 (11.9%) |
| Sector: | | | |
| Public | 112 (63%) | 106 (68%) | 152 (69%) |
| Private | 65 (37%) | 49 (32%) | 67 (31%) |

**Table 2. Factor loading of the Existence Scale items (n = 551).**

| Items | Factor 1 SDi | Factor 2 STr | Factor 3 Fr | Factor 4 Re |
|---|---|---|---|---|
| Q1 | **0.71** | 0.13 | -0.02 | -0.02 |
| Q2 | **0.66** | -0.02 | 0.14 | 0.08 |
| Q3 | **0.63** | 0.15 | 0.02 | 0.11 |
| Q4 | **0.87** | 0.07 | -0.06 | 0.03 |
| Q5 | **0.67** | 0.02 | 0.10 | 0.08 |
| Q6 | **0.67** | 0.21 | 0.11 | 0.16 |
| Q7 | **0.74** | 0.12 | 0.15 | 0.12 |
| Q8 | **0.74** | 0.10 | 0.14 | 0.09 |
| Q9 | 0.13 | **0.75** | -0.02 | 0.14 |
| Q10 | 0.12 | **0.73** | 0.14 | 0.13 |
| Q11 | 0.15 | **0.63** | 0.01 | 0.11 |
| Q12 | 0.02 | **0.57** | 0.10 | 0.23 |
| Q13 | 0.10 | **0.58** | 0.03 | 0.06 |
| Q14 | -0.06 | **0.81** | 0.08 | 0.22 |
| Q15 | 0.02 | **0.69** | 0.02 | 0.14 |
| Q16 | 0.05 | **0.59** | 0.06 | 0.15 |
| Q17 | 0.13 | **0.84** | -0.01 | 0.11 |
| Q18 | 0.12 | **0.81** | 0.12 | 0.20 |
| Q19 | -0.03 | 0.09 | **0.74** | 0.17 |
| Q20 | 0.14 | 0.16 | **0.88** | 0.09 |
| **Q21** | 0.02 | 0.07 | **0.06** | 0.16 |
| Q22 | 0.14 | 0.19 | **0.86** | 0.22 |
| Q23 | 0.01 | 0.09 | **0.72** | 0.27 |
| Q24 | -0.06 | -0.08 | **0.54** | 0.11 |
| Q25 | 0.10 | 0.11 | **0.85** | 0.06 |
| Q26 | 0.01 | 0.14 | **0.80** | 0.18 |
| Q27 | 0.02 | 0.08 | **0.51** | 0.32 |
| Q28 | 0.13 | 0.16 | **0.86** | 0.27 |
| **Q29** | 0.01 | 0.07 | -0.02 | **0.19** |
| Q30 | 0.07 | 0.08 | 0.15 | **0.76** |
| **Q31** | 0.00 | 0.07 | 0.25 | **0.13** |
| Q32 | 0.03 | 0.03 | -0.02 | **0.76** |
| Q33 | 0.04 | 0.09 | 0.01 | **0.76** |
| Q34 | 0.05 | 0.09 | 0.04 | **0.93** |
| Q35 | -0.06 | 0.07 | 0.05 | **0.88** |
| Q36 | 0.00 | 0.04 | 0.00 | **0.87** |
| Q37 | 0.01 | 0.05 | 0.22 | **0.72** |
| **Q38** | 0.0 | 0.12 | 0.20 | **-0.08** |
| Q39 | 0.11 | 0.08 | 0.01 | **0.85** |
| Q40 | 0.08 | -0.04 | 0.05 | **0.90** |
| Q41 | -0.02 | 0.11 | -0.12 | **0.76** |
| **Initial eigenvalues** | **11.55** | **5.06** | **3.96** | **3.27** |
| **Percentages of variance explained** | **28.17** | **12.34** | **9.66** | **7.98** |
| **Cumulative variance** | **28.17** | **40.51** | **50.17** | **58.15** |

SDi: Self-distance. STr: Self-transcendence. Fr: Freedom. Re: Responsibility

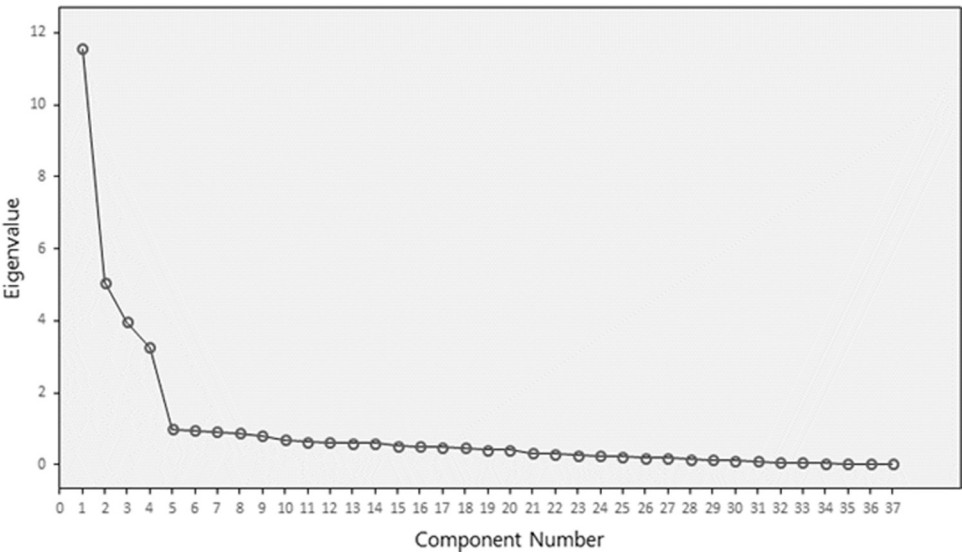

**Fig 1. Scree plot for the translated version of ES (37 items). eigenvalues > 1.5.**

Items of SDi construct have factor loading between (0.63–0.87), explaining 28.17% of the items of variance with eigenvalues more than 1.5. Items of STr construct have factor loading between (0.57–0.84), explaining 12.34% of the item's variance with eigenvalues more than 1.5. Items of Fr construct have factor loading between (0.51–0.88), explaining 9.66% of the variance with eigenvalues more than 1.5. Item 21 in Fr, was omitted due to low factor loading 0.06. Items of Re construct have factor loading between (0.72–0.93), explaining 7.98% of the variance with eigenvalues more than 1.5. Items (Q29, Q31, and Q38) in Re were deleted from the scale due to low factor loading. The translated ES was compromised from 41 items to 37 items. We performed the EFA again with 37 items and reported the output as follows: The KMO test result was 0.90, The Bartlett's test of sphericity was ($\chi2$: 19997.5; df: 666; $p < 0.001$). Factor loadings of SDi construct have factor loading between (0.45–0.78), explaining 28.64% of the items of variance with eigenvalues greater than 1.5. Items of STr construct have factor loading between (0.53–0.98), explaining 13.56% of the item's variance with eigenvalues more than 1.5. Items of Fr construct have factor loading between (0.47–0.94), explaining 10.57% of the item's variance with eigenvalues more than 1.5. Finally, items of Re construct have factor loading between (0.56–0.92), explaining 8.80% of the item's variance with eigenvalues more than 1.5. As a result, 37 items were considered as a reduction of several variables with the total cumulative variance of 61.57% explained and had a better fit regarding scree plot (Fig 1).

## Confirmatory factor analysis

The CFA is used to determine whether the latent variables including self-distance, self-transcendence, freedom, and responsibility adequately fit the data. 37 items were measured to evaluate the factor structure of the ES and its subscales. Fig 2. shows the measurement model.

Fig 2 illustrates the factor loadings of the translated ES items. Self-distance showed a range of loading between 0.56 and 0.88, self-transcendence items loading range was between 0.36 and 1.00, freedom items indicated a range of loading between 0.34 and 1.00, and responsibility items revealed a range of loading between 0.62 and 0.99. However, the fit indices for this measurement model did not reach the criteria set a priori to evaluate the factor structure of the ES

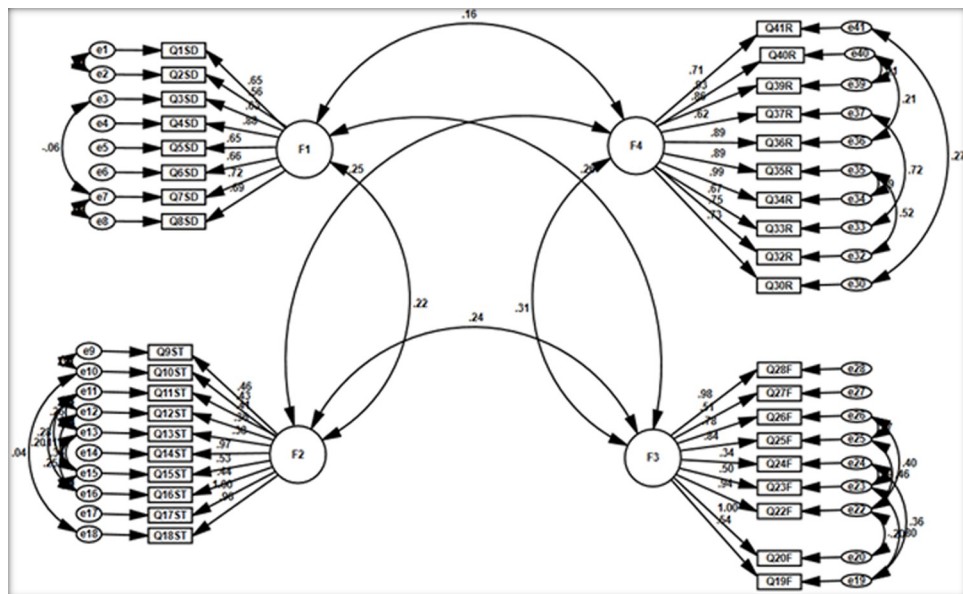

**Fig 2. CFA measurement model 2 for the Existence Scale (AMOS V26).** F1: self-distance, F2: self-transcendence, F3: freedom, F4: responsibility.

scale. Hence, the decision was made to re-specify the model through inspecting the Modification Indices (MI). The MI value above 15 indicated there has been redundancy in the Model [33]. After the redundant items were identified in the measurement model 1, we had to choose between deleting the items or to "free the parameter estimates" to assess the factor structure of the 4 subscale (factor) model. After freeing these parameter estimates, values of the fit indexes had improved in measurement Model 2 as presented in Table 3.

Table 3 presents the fit indexes for measurement Model 1 and 2. Model 1 represents the measurement model without freeing the error estimates. On the other hand, Model 2 represents the model after freeing the error estimates and it has achieved the criteria set a priori regarding the goodness of model fit; relative chi-square is 2.117 which is less than 3, RMSEA is 0.045 indicating an excellent model, CFI values 0.966 exceeding 0.95 and indicating excellent model fit, GFI is 0.882 which is indicative acceptable model, IFI and TLI are greater than 0.95 indicating excellent model fit.

## Convergent and discriminant validity

The results showed that the AVE values for all ES subscales were above 0.50 indicative of convergence. In addition, the CR values were all above 0.70 supporting the convergent validity of all ES subscales. The correlations between the exogenous construct of ES including (SDi, STr,

**Table 3. Fit indexes results for the CFA model of the Existence Scale (n = 551).**

| Models | $\chi2$[a] | d.f.[b] | p[c] | $\chi2$/d.f.[d] | GFI[e] | CFI[f] | IFI[g] | TLI[h] | RMSEA[i] |
|---|---|---|---|---|---|---|---|---|---|
| Model 1 | 2315.36 | 773 | <0.001 | 2.964 | 0.832 | 0.861 | 0.861 | 0.823 | 0.079 |
| Model 2 | 1261.97 | 596 | <0.001 | 2.117 | 0.882 | 0.966 | 0.967 | 0.962 | 0.045 |

Model 1, Model 2: 37 items, χ2: Chia square, d.f.: degree of freedom, GFI: Goodness of Fit Index, CFI: Comparative Fit Index, IFI: Increment Fit Index, TLI: Tucker Lewis Index, RMSEA: Root Mean Square Error of Approximation.[a] Chia square.

**Table 4. ES subscales Cronbach's Alpha, CR, AVE, and bivariate correlations (n = 551).**

| variable | SDi | STr | Fr | Re | cronbach's alpha | CR$^e$ | AVE$^f$ |
|---|---|---|---|---|---|---|---|
| Self-distance | **.85** | | | | 0.88 | 0.88 | 0.73 |
| Self-transcendence | 0.22 | **.84** | | | 0.90 | 0.86 | 0.70 |
| Freedom | 0.26 | 0.24 | **.88** | | 0.90 | 0.91 | 0.77 |
| Responsibility | 0.16 | 0.25 | 0.31 | **.86** | 0.93 | 0.95 | 0.74 |

CR: Composite reliability. AVE: Average variance extracted, SDi: Self-Distance, STr: Self-Transcendence, Fr: Freedom, Re: Responsibility. Diagonal line in bold font presented the square root of AVE.

Fr, and Re) are presented in Table 4. All values of the square root of AVE were higher than the inter-factor correlations supporting the discriminant validity of all ES subscales. Moreover, AVE values of ES (SDi, STr, Fr, and Re) were greater than MSV (0.53, 0.49, 0.59, 0.54) and ASV (0.28, 0.24, 0.34, 0.29), respectively. These values support the discriminant validity of all ES subscales.

## Reliability

The results revealed that the values of the Cronbach's α and CR support the reliability of the translated ES subscales (Table 4).

## Discussion

The current study was conducted to translate the ES and test its psychometric properties in Jordan populations. The results of translation phase and face validity of the translated ES yielded 46 items and none of the items was removed from the original ES. In content validity, four items from self-transcendence construct and one item from freedom construct were omitted based on their low values in Lawshe's CVR Table, leaving the translated ES on 41-items. However, the results of the EFA showed that the items of the translated ES scale load on four distinct factors. The four distinct factors explained more than half of the variance. Due to low factor loading in EFA, three items from responsibility construct and one item from freedom construct were deleted. Collectively, by removing 9 items from the original ES, the total of 37 items has remained in the translated ES.

Regarding the CFA analyses, the modified model achieved the goodness of model fit indices, and the factor structure was then supported of the four-factor model. Finally, the results of the study confirmed the convergent and discriminant validity of the translated ES among Jordan populations.

Comparing our result with previous studies regarding validity of the ES. A study in Russian, aimed to test the ES psychometric properties of Russian version in a sample of general population. The result of this study was similar to our findings regarding translation, face, and content validity; the only difference was deleted 3 items in construct validity due to low factor loading, and thus a Russian version of ES has yielded 43 items [10]. Another study carried out in Iran (2018) among 500 teachers. They found that the ES achieved criteria validity, with four factors loading with an eigenvalue more than one, and the CFA was supportive of the four-factor model. In the EFA, the Iranian ES was reduced to 38 items (8 items were deleted from the origin ES due to low factor loading) through four constructs [2]. The results reported in the current study are in line with those of previous studies regarding face validity, translation phase, content validity, and CFA. It is noteworthy to mention that other study has measured the psychometric properties of the ES based on four samples; school teachers, school

principals, pastors, and social workers in Netherlands. The results of CFA and construct validity did not confirm the validity of the ES and, hence, the translated ES in Netherlands is not valid to measure the existential fulfillment [11]. There are several reasons regarding the Netherlands ES did not confirm the validity, it seems that the construct of ES and the translation process had been formulated improperly or may be referred that several items of the Netherlands ES were ambiguously worded.

Regarding the reliability phase of the translated ES, the values of the Cronbach's α and CR supported the internal consistency of the four subscales. The current study finding agrees with those of previous studies [2, 10], by showing that Cronbach alpha and CR values as well as test-retest reliability coefficient having good reliability.

Comparing our translated ES with another similar tools. A study conducted in Iran among 1210 students to assess the validity and reliability of the self-transcendence scale (STS), which is the main concept in the existential theory. After examined the translation phase, face validity, construct and discriminant validity, and internal reliability consistency, the result showed that the STS is found to be acceptable regarding validity and reliability and it can be used to measure self-transcendence [34]. A study on Norwegian nurses revealed that the purpose in life scale is valid and reliable to measure the existential meaning of life since the content, EFA, CFA, CR and Cronbach's alpha were acceptable [35]. Another study among Hong Kong Chines students showed that the meaning in life scale is positively associated with life satisfaction scale and it is a valid tool to measure the meaning in life to achieve the construct, CFA, and reliability criteria [36].

Having found an adequate psychometric property of the translated ES, we have concluded that it is suitable for clarifying the purpose and meaning in life among nurses, teachers, and students due to the stressful nature of their work. Shortening of the original ES from 46 to 37 items is one of our important findings. The study findings should be interpreted within the context of certain limitations. The participants were recruited using non-probability sampling method. Therefore, the generalizability of the results could be limited. Moreover, the sample population consisted of three professions, so we ran EFA and CFA on the same dataset (551 participants) to overfit our results.

## Conclusion

This study is one of the first investigations to assess the psychometric properties of the translated ES among Jordan populations. The translated ES is a valid and reliable scale since it was achieved all psychometric properties criteria. It can be used to measure the life meaning and purpose and to assess the existential fulfillment among the recruited areas. Future studies are therefore recommended to apply the translated ES in different Arab countries to confirm and validate these findings.

## Acknowledgments

The authors would like to thank all participants for their valuable contribution in this study.

## Author Contributions

**Conceptualization:** Othman A. Alfuqaha, Mohammed M. Al-Hammouri.

**Data curation:** Othman A. Alfuqaha, Mohammed M. Al-Hammouri, Jehad A. Rababah, Bayan A. Alfoqha, Ola N. Alfuqaha, Moh'd Fayeq F. Haha, Suzan S. Musa, Aseel A. Matter.

**Formal analysis:** Jehad A. Rababah, Bayan A. Alfoqha.

**Investigation:** Mohammed M. Al-Hammouri.

**Software:** Ola N. Alfuqaha.

**Supervision:** Othman A. Alfuqaha.

**Validation:** Othman A. Alfuqaha, Jehad A. Rababah.

**Visualization:** Othman A. Alfuqaha, Mohammed M. Al-Hammouri.

**Writing – original draft:** Othman A. Alfuqaha.

**Writing – review & editing:** Othman A. Alfuqaha, Mohammed M. Al-Hammouri, Jehad A. Rababah, Bayan A. Alfoqha, Ola N. Alfuqaha, Moh'd Fayeq F. Haha, Suzan S. Musa, Aseel A. Matter.

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
