## [Decision Letter · Decision Letter 0]

31 Jan 2022

PONE-D-21-27986Psychometric Properties of the Arabic Version of the Existence ScalePLOS ONE

Dear Dr.Alfuqaha,

Thank you for submitting your manuscript to PLOS ONE. After careful consideration, we feel that it has merit but does not fully meet PLOS ONE’s publication criteria as it currently stands. Therefore, we invite you to submit a revised version of the manuscript that addresses the points raised during the review process.

We look forward to receiving your revised manuscript.

Kind regards,

Ali Montazeri

Academic Editor

PLOS ONE

Journal Requirements:

Reviewers' comments:

Reviewer's Responses to Questions

**Comments to the Author**

1. Is the manuscript technically sound, and do the data support the conclusions?

Reviewer #1: Partly

Reviewer #2: Yes

2. Has the statistical analysis been performed appropriately and rigorously? 

Reviewer #1: No

Reviewer #2: Yes

3. Have the authors made all data underlying the findings in their manuscript fully available?

Reviewer #1: No

Reviewer #2: Yes

4. Is the manuscript presented in an intelligible fashion and written in standard English?

Reviewer #1: Yes

Reviewer #2: Yes

5. Review Comments to the Author

Reviewer #1: Psychometric Properties of the Arabic Version of the Existence Scale

Thank you for the opportunity to review the manuscript. I reviewed this manuscript carefully with a great interest. I respectfully provided my comments below

1: There is also a Hungarian version of this questionnaire

Materials and methods

1:Separate the description of the original questionnaire from the translation process.

Page 6, paragraph 3, line 8

Are you allowed to delete the question on content validity without the consent of the questionnaire designer? Because this is a psychometer, so you can only do quality content and face validation

3:The choice of PCA to evaluate the factor structure of the Existence Scale appears somewhat rudimentary given the increasingly sophisticated statistically methods that have been used to evaluate the scale since the publication of the original article. It would be good to know why the author has chosen to use PCA rather than using structural equation models which have become much more commonly used to test the psychometric properties of scales. Maximum likelihood should be used

4: In convergent and discriminant validity, please report ASV, MSV.

5: Mention the reason you did not report the ICC for stability?

Result

The results need to be re-examined based on the suggestions I made in the method section.

Figure 1. Scree plot, show 5 factor

Figure 2. CFA measurement model 2 for the Existence Scale, Questions 24. 9, 10, 11, 12, 13 and 16 should be removed from the model due to an operating load of less than 0.5. So the CFA has to be done again after the questions are removed

Reviewer #2: 1. After performing the exploratory factor analysis and concluding to remove 4 items, it is necessary to perform this analysis again with 37 items and report the output table of the relevant information.

2. Regarding convergent validity, the authors write: "Convergent and discriminant validity were evaluated by examining the bivariate correlations between the translated ES and comparing the correlations with the squared root of the average variance extracted (AVE)." This statement is general and ambiguous regarding the study of two differential and convergent validities. Please be documented and explained in more clarity and detail.

There was a special reason for dividing the age of the participants into two groups, under 35 and over 35 years old, in the demographic characteristics report section.4. Existence questionnaire is related to which range of population? What was the reason for selecting the sample population [the three samples consisted of three professions (nursing, teaching, and students)] in the factor analysis stage of this study?

5.Number of sample population to perform exploratory and confirmatory factor analysis based on what criteria (mention the relevant reference).

6. Considering the recommendation of some experts to differentiate the two sample populations for two exploratory and confirmatory factor analyzes, has this criterion been considered in the present study?

6. PLOS authors have the option to publish the peer review history of their article (what does this mean?). If published, this will include your full peer review and any attached files.

Reviewer #1: **Yes: **Razieh Bandari

Reviewer #2: No

---

## [Author Response · Author response to Decision Letter 0]

23 Feb 2022

Response to Reviewers

Dear editor,

We would like to thank you and both reviewers for their valuable comments and feedbacks. Point-by-point responses to reviewers are listed below.

Reviewer #1: Psychometric Properties of the Arabic Version of the Existence Scale

Thank you for the opportunity to review the manuscript. I reviewed this manuscript carefully with a great interest. I respectfully provided my comments below.

Comment 1:

1: There is also a Hungarian version of this questionnaire

Response 1:

Thank you for your valuable comments. In response to this comment, we added a Hungarian version of ES in the introduction section.

Comment 2:

Materials and methods

1: Separate the description of the original questionnaire from the translation process.

Response 2:

Done. Please see our revised manuscript.

Comment 3:

Page 6, paragraph 3, line 8

Are you allowed to delete the question on content validity without the consent of the questionnaire designer? Because this is a psychometer, so you can only do quality content and face validation

Response 3:

Thank you for your valuable comments. We took the official permission from Prof. Dr.med.Dr.phil. Alfried Längle to translate, omit, and modify the scale as required. 

Comment 4:

3:The choice of PCA to evaluate the factor structure of the Existence Scale appears somewhat rudimentary given the increasingly sophisticated statistically methods that have been used to evaluate the scale since the publication of the original article. It would be good to know why the author has chosen to use PCA rather than using structural equation models which have become much more commonly used to test the psychometric properties of scales. Maximum likelihood should be used

Response 4:

Thank you for your advice. We completely agree with you regarding using structure equation models have become commonly used among researchers. The PCA is still a good choice for researchers either as in your fantastic paper. “Psychometric Properties of the Persian Version of the Quality of Life in Early Old Age (CASP- 19)”. However, we have used PCA as we believe that there is redundancy among ES items and are relatively measuring the same construct. Because of this redundancy, we believe that it should be possible to reduce the observed variables into a smaller number of principal components that will account for most of the variance in the observed variables. Regarding Maximum likelihood, we depend on it in our result in AMOS software. However, we added the Maximum likelihood in our revised manuscript under confirmatory factor analysis Page 9, Paragraph 1. 

Comment 5:

4: In convergent and discriminant validity, please report ASV, MSV.

Response 5:

Done as your suggestion. Please see our revised manuscript.

Comment 6:

5: Mention the reason you did not report the ICC for stability?

Response 6:

We believe that the Cronbach alpha and composite reliability are sufficient for stability. Moreover, ICC values are somewhat sensitive to subject variability, which could lead to different values even for the same measurement errors in similar dimensions.

Comment 7:

Result

The results need to be re-examined based on the suggestions I made in the method section.

Response 7:

Done

Comment 8:

Figure 1. Scree plot, show 5 factor

Response 8:

Thanks for pointing this issue. In fact, the factor number 5 loaded at 1.025. So we decided to consider Eigenvalues equal or above 1.5. However, we re-write the cut-off-point regarding this. Please see our revised manuscript under “construct validity”. 

Comment 9:

Figure 2. CFA measurement model 2 for the Existence Scale, Questions 24. 9, 10, 11, 12, 13 and 16 should be removed from the model due to an operating load of less than 0.5. So the CFA has to be done again after the questions are removed

Response 9:

Thanks for your advice. We totally agree with you that factor loadings of 0.5 and higher will represent higher contributions and be more practical. But we found that the cutoff value of 0.30 and greater also represent a fair contribution in latent variables. We added this cutoff point value in our revised manuscript under Confirmatory Factor Analysis”.

Here are some references to support our evidence.

- Hoyle HR. Confirmatory Factor Analysis. Handbook of Applied Multivariate Statistics and Mathematical Modeling. 2000. P:173-174.

- Yusoff MS, Rahim AF, Mat Pa MN, See CM, Ja'afar R, Esa AR. The validity and reliability of the USM Emotional Quotient Inventory (USMEQ-i): its use to measure Emotional Quotient (EQ) of future medical students. International Medical Journal. 2011 Dec 1;18(4):293-9.

Reviewer #2: 

Comment 1:

1. After performing the exploratory factor analysis and concluding to remove 4 items, it is necessary to perform this analysis again with 37 items and report the output table of the relevant information.

Response 1:

Thanks for this comment, therefore, we performed EFA again with 37 items and reported the results in our revised manuscript as your suggestion. Page 12

Comment 2:

2. Regarding convergent validity, the authors write: "Convergent and discriminant validity were evaluated by examining the bivariate correlations between the translated ES and comparing the correlations with the squared root of the average variance extracted (AVE)." This statement is general and ambiguous regarding the study of two differential and convergent validities. Please be documented and explained in more clarity and detail.

Response 2:

As suggested by the reviewer, we paraphrased this sentence to be clear for audience. A new paragraph was added as follows: “Convergent validity was evaluated by examining average variance extracted (AVE). The AVE had to be greater than 0.50 [30]. Discriminant validity was assessed by examining squared root of the AVE, maximum shared variance (MSV), and average shared squared variance (ASV). The square root of AVE should be greater than inter-factor correlations of ES (SDi, STr, Fr, and Re) [30]. Both MSV and ASV values had to be lower than AVE values [27].”

Comment 3:

There was a special reason for dividing the age of the participants into two groups, under 35 and over 35 years old, in the demographic characteristics report section

Response 3:

In the near future, we will use our data to investigate the existential vacuum among participants, therefore, we think the title of our next manuscript is “The role of socio-demographic factors with the existential vacuum. As you noticed in our paper, several demographic factors were included such as: gender (male vs female) age (under 35 and over 35) years old …. etc. Out of the box, is it a question of whether people over 35 are fulfilled in their lives or the other way around? 

Comment 4:

4. Existence questionnaire is related to which range of population? What was the reason for selecting the sample population [the three samples consisted of three professions (nursing, teaching, and students)] in the factor analysis stage of this study?

Response 4:

The reason for selecting three sample is mentioned in our study under the Introduction section as:

“In this regard, nursing, teaching, and student professions are considered stressful professions that lead to much psychological distress such as burnout, sense of emptiness, low self-evaluation, and negative affect of meaning in life. This makes them at high risk for existential vacuum [14-16]”.

“Of note, the meaning in life among social professions (i.e., nursing) that have extensive contact with people not only affects the professionals themselves but is also affected the quality-of-life people [2, 3]”.

“The recruited areas (nursing, teaching, and students) were selected based on the evidence that experience extensive contact with people and high level of stress [11]. In addition, the recruited samples are expected to improve the generalizability of the findings”.

And also we mentioned the reason under method section “construct validity” as: 

“However, the three samples consisted of three professions (nursing, teaching, and students) that require social occupations, extensive contact with people, and more strenuous job than others”.

However, previous studies also used three samples to explore the validity and reliability of ES. 

11. Brouwers A, Tomic W. Factorial Structure of the Existence Scale. J Artic Support Null Hypothesis. 2012;8(2):21-30.

17. Thege BK, Martos T. Reliability and validity of the Shortened Hungarian Version of the Existence Scale. Körper mit Psyche. 2008;28(1):88-93.

Comment 5:

5.Number of sample population to perform exploratory and confirmatory factor analysis based on what criteria (mention the relevant reference).

Response 5:

Thanks for pointing this issue. In response to this comment a new paragraph was added in our revised manuscript as follows “We selected the three samples population based on previous studies [11, 17] to perform EFA and confirmatory factor analysis (CFA).”

Comment 6:

6. Considering the recommendation of some experts to differentiate the two sample populations for two exploratory and confirmatory factor analyzes, has this criterion been considered in the present study?

Response 6:

No, we did not split our data into two groups, one for EFA and one for CFA. There are many concerns prevented us to do this: (1) refrain from performing EFA and CFA on the same dataset as this yields’ high danger of overfitting. (2) refrain EFA prevented us to assess the internal structure of the Arabic ES scale. (3) Finally, the sample size in our study was divided into 3 major groups and was not large enough (112 nurses, 106 school-teacher, 152 student) to split the dataset into two major groups. 

Your valuable comment was sufficiently addressed in our limitation under the ‘Discussion section’. For more details, please read this article: Fokkema M, Greiff S. How performing PCA and CFA on the same data equals trouble. https://doi.org/10.1027/1015-5759/a000460

In response to this comment, a new paragraph was added in the limitation section: 

“The sample population consisted of three professions, so we ran EFA and CFA on the same dataset (551 participants) to overfit our results”.

We hope now that our revised manuscript is acceptable for publication.

---

## [Editor Report · Decision Letter 1]

1 Apr 2022

PONE-D-21-27986R1Psychometric Properties of the Arabic Version of the Existence ScalePLOS ONE

Dear Dr. Alfuqoha,

Thank you for submitting your manuscript to PLOS ONE. After careful consideration, we feel that it has merit but does not fully meet PLOS ONE’s publication criteria as it currently stands. Therefore, we invite you to submit a revised version of the manuscript that addresses the points raised during the review process.

We look forward to receiving your revised manuscript.

Kind regards,

Ali Montazeri

Academic Editor

PLOS ONE

Journal Requirements:

Additional Editor Comments:

Please clarify in the Methods that you were allowed to omit or modify the original questionnaire with permission form the developers (please include names)

---

## [Author Response · Author response to Decision Letter 1]

2 Apr 2022

Response to Reviewers

Dear editor,

We would like to thank you and for your valuable comments and feedbacks. Point-by-point responses to reviewers are listed below.

Journal Requirements:

Comment 1:

Response 1:

Thank you for your valuable comments. We make sure that all our references are complete and correct.

Comment 2:

Additional Editor Comments:

Please clarify in the Methods that you were allowed to omit or modify the original questionnaire with permission form the developers (please include names).

Response 2:

We appreciate your valuable comments. We added a new statement regarding the permission from Alfried Längle to translate, omit, and modify the existence scale as follows:

“The official permission was sought from the developer (Alfried Längle) to translate, modify, and omit the ES as needed”.

We hope now that our revised manuscript is acceptable for publication.

---

## [Editor Report · Decision Letter 2]

7 Apr 2022

Psychometric Properties of the Arabic Version of the Existence Scale

PONE-D-21-27986R2

Dear Dr. Alfuqaha,

We’re pleased to inform you that your manuscript has been judged scientifically suitable for publication and will be formally accepted for publication once it meets all outstanding technical requirements.

Kind regards,

Ali Montazeri

Academic Editor

PLOS ONE
---

## [Editor Report · Acceptance letter]

8 Apr 2022

PONE-D-21-27986R2 

Psychometric Properties of the Arabic Version of the Existence Scale 

Dear Dr. Alfuqaha:

I'm pleased to inform you that your manuscript has been deemed suitable for publication in PLOS ONE. Congratulations! Your manuscript is now with our production department. 

Kind regards, 

on behalf of

Professor Ali Montazeri 

Academic Editor

PLOS ONE